# Harnessing the Gut Microbiome: To What Extent Can Pre-/Probiotics Alleviate Immune Activation in Autism Spectrum Disorder?

**DOI:** 10.3390/nu16152382

**Published:** 2024-07-23

**Authors:** Yuqi Wu, Qi Su

**Affiliations:** 1Microbiota I-Center (MagIC), Hong Kong SAR, China; 1155201694@link.cuhk.edu.hk; 2Department of Medicine and Therapeutics, The Chinese University of Hong Kong, Hong Kong SAR, China

**Keywords:** autism spectrum disorder, pre-/probiotics, gut–brain axis, immune response, healthcare

## Abstract

Children diagnosed with autism spectrum disorder (ASD) are at an increased risk of experiencing gastrointestinal (GI) discomfort, which has been linked to dysfunctions in the microbiome–gut–brain axis. The bidirectional communication between gut and brain plays a crucial role in the overall health of individuals, and alterations in the gut microbiome can contribute to immune activation and gut–brain dysfunction in ASD. Despite the limited and controversial results of pre-/probiotic applications in ASD, this review comprehensively maps the association between ASD clinical symptoms and specific bacterial taxa and evaluates the efficacy of pre-/probiotics in modulating microbiota composition, reducing inflammatory biomarkers, alleviating difficulties in GI distress, sleep problems, core and other ASD-associated symptoms, as well as relieving parental concerns, separately, in individuals with ASD. Beyond simply targeting core ASD symptoms, this review highlights the potential of pre-/probiotic supplementations as a strategy to modulate gut homeostasis and immune response, and to delineate the potential mechanisms by which its direct or mediating effects can alleviate gut–brain dysfunction and poor nutritional status in ASD management. Further well-designed randomized controlled trials are needed to strengthen the existing evidence and establish optimal protocols for the use of pre-/probiotics in the context of ASD.

## 1. Introduction

With a global prevalence impacting approximately 78 million individuals, autism spectrum disorder (ASD) imposes a significant epidemiological and societal burden worldwide [1]. ASD is characterized by difficulties in social communication/interaction, as well as repetitive/restricted behaviors and interests, which emerge early in neurodevelopment.

Children diagnosed with ASD are at a four-fold elevated risk of suffering gastrointestinal (GI) discomfort, including constipation, changes in appetite, abdominal pain, and bloating, which are aligned with the severity of ASD [2,3]. The concept of the gut–brain axis refers to the crosstalk between gut microbiota and neurological signals, crucial for maintaining gut microbial homeostasis and brain function [4,5]. Increased gut permeability, often termed “leaky gut”, is frequently observed in children with ASD and is believed to result from immune activation in the host [4,5]. Moreover, among those children, microbial community changes promote the growth of *Candida* spp., leading to their dominance and hindrance of commensal microbe recolonization, reflecting the pivotal role of the gut microbiome in immune responses [6]. Of note, immune abnormalities are commonly reported in individuals with many medical conditions that are overrepresented in ASD, including autoimmune allergy, clinical immunodeficiency, GI symptoms, intellectual disability, and attention-deficit/hyperactivity disorder (ADHD) [4,7,8]. Immune signals transmitted to the brain through humoral and neural pathways potentially interplay with the gut microbiome, leading to the synthesis of inflammatory mediators [7,8]. This process may further exacerbate autistic-like behaviors and related symptoms [7,8]. Hence, understanding how gut microbiome-mediated immune dysfunction influences the health of ASD children will facilitate identifying those at higher risk and the subsequent development of targeted strategies for treatment. The ultimate goal is to improve overall functioning and quality of life for individuals living with ASD.

A growing body of evidence suggests a close relationship between diet and ASD development, which partially engages in nutrient metabolism pathways and gut microbes [8,9,10,11,12]. Given the general features of selective eating in children with ASD, these children often experience nutritional deficiencies, which in turn can have adverse effects on brain function and development [12]. In the Chinese population, Liu et al. [13] found a higher proportion of children with ASD exhibiting food avoidance, severe eating problems, and constipation, accompanied by a higher risk of Vitamin A deficiency compared to their typically developing (TD) peers. In the same study, more vitamin and mineral insufficiencies were observed in children with ASD than without, and their serum concentrations were associated with levels in the Autism Behavior Checklist (ABC) and Social Responsiveness Scale (SRS), especially for Vitamin A [13,14]. Intriguingly, in addition to lower levels of serum retinol and 25-OH Vitamin D observed in children with picky eating, new food resistance, and severe ASD symptoms, the deficiencies of retinol and 25-OH Vitamin D exhibited joint effects on ASD rating scale, sensory, social withdrawal, and stereotypic behavior when they did not meet sufficient levels [15]. Thereby, concurrent immune activation and atypical eating behaviors can lead to imbalanced dietary intake and malnutrition, creating a vicious cycle that affects the control of ASD symptoms. Unfortunately, there is currently a lack of a comprehensive view and strategy targeting gut microbiome and immune regulation to manage ASD-related symptoms and nutritional problems in the clinical setting.

Despite ongoing debates regarding the direct causal relationship between the gut microbiome and ASD [7,8,16], emerging evidence suggests that the gut microbiome may serve as a crucial mediator in the context of ASD-induced dietary imbalances and nutritional status, closely linked to immune responses [8,17,18]. This implies that modulating the gut microbiome through interventions such as pre-/probiotics could offer a promising strategy for improving immune function and nutritional status in ASD individuals [18]. In addition to the primary focus on addressing core ASD symptoms, this review aims to highlight the potential of pre-/probiotics in modulating microbiota composition, reducing inflammatory biomarkers, alleviating difficulties in GI issues, sleep patterns, and other ASD-associated symptoms, as well as addressing parental concerns. By gaining a deeper understanding of the intricate relationships between the gut microbiome, immune activation, and ASD pathology, researchers and healthcare providers can develop more effective and targeted strategies to harness the potential of pre-/probiotics in promoting gut–brain health and overall well-being in the management of ASD.

## 2. Gut Microbiome-Mediated Immune Dysregulation Co-Occurs with Poor Nutritional Status

Immune function is generally classified into two types: innate and adaptive [11]. The innate immune response serves as a nonspecific defense system against pathogens, while the adaptive process is highly specific and capable of recognizing previously encountered pathogens. Emerging evidence has suggested that individuals with ASD were more likely to exhibit abnormalities in both types of immune activity, mirroring the high co-occurrences of GI discomfort and eating problems observed in this population [7,8,11,18,19]. In terms of these comorbidities, dysregulation of the immune system in autism may further contribute to an imbalanced diet and poor nutritional status. Specifically, those with ASD frequently experience dysregulation of autoimmunity in both the brain and peripheral immune system, characterized by elevated levels of inflammatory cytokines and activation of immune cells [19]. Importantly, commensal microbes play a role in communicating with the brain and participate in immune regulation through the production of certain metabolites, such as short-chain fatty acids (SCFAs), underscoring the gut microbiome as a promising target for maintaining gut-immune homeostasis [10,19]. Taken as a whole, the poor nutritional profile may not just be a stimulus for exaggerated autistic traits, but rather a health outcome accompanying the immune dysregulation mediated by the gut microbiome [12]. This could be manifested through the aspects of gut dysbiosis, gastrointestinal symptoms, food allergy and digestive discomfort, as well as neuroinflammation (Figure 1).

### 2.1. Gut Dysbiosis

Elevated gut permeability observed in children with ASD may result from immune activation in the GI tract [5,20,21]. For example, a high abundance of *Clostridium* spp., and the decreased alpha diversity and levels of *Bacteroides*, *Bifidobacterium* spp., and *Lactobacillus* spp., may promote a pro-inflammatory environment, exacerbating the immune dysfunction characteristic of autism [5]. The symbiotic relationship between gut dysbiosis and its host presents multiple challenges to clinical healthcare in children with ASD, affecting metabolic disturbances, immune activation, nervous system dysfunction, and dysregulated eating behaviors [12,21]. Further, based on this crosstalk between immune activation and gut dysbiosis, the proposed food reward system can be mediated by neurotransmitters, bioactive lipids, bacterial metabolites, and hormones released from the gut, leading to disordered eating behaviors [22]. Consequently, these comorbid immune activations and atypical eating behaviors may incur an imbalanced dietary intake and malnutrition, which, in turn, forms a vicious cycle affecting the control of autistic traits [12].

In the nervous pathway, gut microbes have interactions with key neurotransmitters, like γ-aminobutyric acid (GABA), dopamine, and serotonin, as well as communication with the vagus nerve [22]. The role of GABA in mediating autism through the gut–brain axis should be highlighted, as lower levels of GABA relative to glutamate were inversely correlated with the severity of social dysfunction and sensory abnormalities in children with ASD [23]. The imbalanced levels of plasma glutamic acid and glutamine found in autistic individuals also mirrored this observation [24]. Although the hypothesis concerning the abnormalities of circulating GABA and certain amino acids has arisen from metabolic disruptions, it is also suggested that these amino acid deficiencies stem, at least in part, from poor protein nutrition and selective eating habits [24,25]. Moreover, endocannabinoids, a class of lipid-based neurotransmitters that will be discussed below, also play a role in the gut–brain interaction [26]. Studies have shown that the endocannabinoid system is implicated in ASD, with findings of reduced anandamide and other endocannabinoids, as well as altered expressions in cannabinoid receptors and metabolic enzymes [26,27]. In this regard, endocannabinoids may serve as a novel therapeutic target to modulate the atypical eating patterns and food reward process through the gut–brain axis [26]. Upon examining the research findings which highlighted the key roles of the serotonin system in autism, there are evident overlaps among elevated levels of blood serotonin, sensory dysfunction, tryptophan depletion, and GI distress [28]. This can be explained by the fact that serotonin may directly regulate secretory and sensory–motor functions within the GI tract [29]. The alteration of serotonin is involved in immune cell function and angiogenesis during gut inflammation in the meantime [29]. To be specific, individuals predisposed to eating disorders exhibited increased extracellular serotonin levels, which was attributed to the intensive satiety and potential interaction between serotonin and dopamine [30]. These findings provide valuable insight into understanding the role of the serotonin system in the gut–brain axis and its relevance to eating behaviors, which can guide the treatment of ASD-related symptoms.

On the other hand, in the systemic pathway, the interaction between microbial metabolites, gut hormones, and nutrients with the vagus nerve facilitates the crosstalk between gut dysbiosis and immune dysregulation in the central nervous system (CNS) [11]. In ASD, elevated proinflammatory microbial metabolites, along with sustained dysbiosis marked by reduced *Lactobacillus* spp. and diminished *Clostridium* spp. (including the absence of *C. difficile*), have been significantly associated with greater autistic severity [31]. Additionally, the presence of *Candida* spp. may contribute to the malabsorption of minerals and carbohydrates, potentially impacting the nutritional status and playing a role in the pathophysiology of ASD [31]. The gut microbiota has been shown to participate in the synthesis of satiety hormones, such as peptide YY (PYY) and glucagon-like peptide-1 (GLP-1). Notably, SCFAs, particularly butyrate, have a profound impact on this process. In addition to enhancing CNS function by inhibiting histone deacetylases to prevent neural dysfunction, butyrate also promoted the secretions of GLP-1 and PYY, which could influence both homeostatic and hedonic food intakes [22]. Furthermore, the evidence demonstrated that maternal immune activation, caused by factors such as infection, stress, and autoimmune diseases, acts as a driver in GI barrier defects, gut dysbiosis, and more severe autistic traits in early life [5,11,32]. However, the specific contributions of the child’s immune activation versus maternal immune activation to behavioral modulation in offspring remain unknown.

### 2.2. Gastrointestinal Symptoms

It is important to note that nearly 70% of children with ASD experience GI problems [33]. The strong correlation between GI scores and the Autism Treatment Evaluation Checklist (ATEC), along with its subscales, suggested a significant association between GI issues and ASD symptoms [34]. As previous research stated, over 56% of functional GI disorders occurred before the onset of eating disorders [35]. Avoiding certain foods due to food intolerance, such as eliminating lactose-containing milk from the diet due to lactose intolerance, may also pose nutritional challenges. It is crucial to recognize that disordered eating behaviors are not sufficient to develop a diagnosed eating disorder, but all individuals with eating disorders engage in disordered eating, which can result in malnutrition. Regardless of the severity, ranging from mild avoidance to a diagnosed disorder, these eating difficulties are highly prevalent among individuals with GI issues [36]. There has been increasing interest in understanding the etiology of GI symptoms related to food avoidance and other atypical eating behaviors under poor GI conditions, considering their contribution to compromised nutritional profiles [21,36]. However, certain aspects of research among children with ASD remain unclear.

One of the common GI challenges, inflammatory bowel disease (IBD), may lead to malabsorption and malnutrition due to chronic inflammation and increased permeability of the GI tract. As a result, those patients may be advised to follow a low-fiber/low-residue diet and limit the consumption of trigger foods to mitigate the severity of symptoms. Medications can also be prescribed to regulate the immune system and alleviate inflammation [35,37]. However, there is evidence suggesting a reciprocal relationship between anorexia nervosa (AN) and IBD, with both conditions influencing the development of eating disorders [35]. The neglected functional GI disorders may distract attention away from the pathology of atypical eating, proceeding poor nutritional status and disordered eating behaviors [35,38]. More notably, given the high prevalence of ASD in children with eating disorders (8–55%), avoidant/restrictive eating behaviors, which frequently co-occur with GI symptoms and ASD, may contribute to the complex clinical manifestations, missed diagnoses, nutritional problems, and developmental deficits [39,40]. Although a recent large-scale prospective study provided comprehensive evidence of relatively high risks of GI disorders, endocrine/metabolic issues, neurodevelopmental disorders (particularly ASD), and allergic disorders in children with avoidant/restrictive eating behaviors, determining the temporal ordering and causality of these associations poses challenges [39]. Therefore, there is a call for additional research to explore the underlying mechanisms of these interrelationships.

### 2.3. Food Allergy and Digestive Discomfort

In line with previous research, a recent large-scale longitudinal study highlighted increased risks of allergic conditions, such as food allergy (HR (hazard ratio), 95% CI (confidence interval): 1.56, 1.32–1.83) and digestive issues (HR, 95% CI: 2.03, 1.76–2.35), in children with avoidant restrictive food intake problems compared to healthy controls. Additionally, these children exhibited a higher likelihood of co-occurring ASD (HR, 95% CI: 9.7, 7.5–12.5) [39]. Moreover, individuals with food allergy and digestive discomfort may encounter disruptions in their gut microbiota composition, resulting in dysbiosis—a state of imbalance within the gut microbial community.

Studies have indicated that individuals with food allergies display variations in the diversity and abundance of gut bacteria compared to non-allergic individuals, which may involve a detrimental cycle that exacerbates symptoms associated with ASD [41]. The gut–brain axis, a bidirectional communication network, has been implicated in the development and manifestation of ASD [4,5]. Moreover, food allergy can significantly shift an individual’s dietary choices and nutrient intake, leading to dietary imbalances, deficiencies in essential nutrients, and alterations in gut microbiota composition [42]. As an autoimmune disorder, food allergy can also trigger immune responses and inflammation in the gut, evidenced by a close connection between autoimmune/inflammatory diseases and a high prevalence of disordered eating behaviors [21]. Therefore, understanding how food allergy influences dietary imbalances and gut microbiome composition is crucial for addressing the complex interactions that may underlie the relationship between food allergy and ASD.

On the other hand, digestive problems commonly experienced by children with ASD can also be implicated in gut microbial dysfunction and immune activation. The gut–brain connection reveals that the efferent signals transmitted through the vagus nerve can modulate the secretion of digestive enzymes in the GI tract, potentially influencing nutrient absorption [11]. Moreover, GI issues often arise from food sensitivities due to heightened sensitivity to normal digestive processes [36], which partially explains the higher prevalence of deficiencies of digestive enzymes and food sensitivities among ASD children. Consequently, insufficient enzyme activity can lead to malabsorption, intestinal inflammation, and associated health challenges [8].

Individuals with AN may have a compromised integrity of the gut mucosal layer due to an elevated presence of specific bacteria that preferentially digest proteins, such as mucin, resulting in increased permeability of the colon [21,43]. Of note, one study involving 101 children with ASD found that a three-month treatment with digestive enzyme therapy significantly alleviated behavioral and GI symptoms, particularly in the domains of socialization, hyperactivity, dietary diversity, and stool consistency [44]. Upon closer examination of the connections between microbial functions and ASD symptoms, Chen et al. [7] observed a positive correlation between social communication/emotion skills and the digestive function in autism. Conversely, a negative association between aggressive/anxiety behaviors and mineral absorption has been noted [7]. Dietary soluble fibers play a crucial role in the human gut, as they are fermented by gut bacteria, contributing to the production of SCFAs. These SCFAs, converted by intestinal *Firmicutes*, *Bacteroidetes*, and *Clostridial*, could further mediate the stabilization of hypoxia-inducible factor-1α (HIF-1α), which is reported to regulate the link between the microbiome homeostasis and the host immune system [45]. However, in some cases, the fermentation process can stimulate intestinal contractions and food intolerance, resulting in gut bloating or discomfort and poorer dietary problems [46]. In conclusion, it is crucial to further investigate the effects of food allergy, digestive problems, and dietary imbalances on the gut microbiome and their potential implications for individuals with ASD. Recognizing the complex interplay among these factors is essential for developing precise interventions to support the physical and mental well-being of the autism community.

### 2.4. Neuroinflammation

Cytokines, which are signaling molecules of the immune system, can be produced directly in the fetal brain or cross an immature blood–brain barrier (BBB). Abnormal levels of cytokines in the brain can disrupt the development of the CNS, leading to behavioral abnormalities related to sociality, selective attention, exploratory inclination, and working memory, which resemble behavioral characteristics of ASD [5,32]. Microglia, a type of macrophages within the CNS, are recognized as key players in this process [19]. Post-mortem examinations of autistic individuals have uncovered elevated microglial cell activity and notable alterations in brain structure, indicating a correlation between ASD and neuronal immune activation [5,19,47]. Furthermore, neuroinflammation may disrupt synaptic function and trigger arginine vasopressin secretion, which has been considered as a biomarker and influence factor in the social behavior of autism [47].

In light of the fact that lipopolysaccharide (LPS) can activate pro-inflammatory pathways, primarily via Toll-like receptor 4 (TLR4), which may induce neuroinflammation in the hypothalamus and downstream homeostatic food intake preferences [22,26], a recent review has proposed insightful connections between the gut microbiome, neuroinflammation, and the pathology of eating disorders, specifically highlighting the role of microglia inflammation in the cause of AN [21]. It is well-known that the gut microbiota and the food reward system are interconnected through endocannabinoids, which primarily derive from n-6 polyunsaturated fatty acids (PUFA). Endocannabinoids, binding to cannabinoid receptors 1 (CB1) and 2 (CB2) in the brain and gut, are recognized as crucial neurotransmitters in the food reward system. In addition to the direct effects, certain proinflammatory derivates of n-6 PUFA may disrupt microglial activity and contribute to neuroinflammation, potentially exacerbating issues related to food avoidance [21,48]. However, these preclinical observations require further validation by understanding the underlying mechanisms of atypical eating behaviors and preferences involving the gut microbiome-mediated neuroinflammation.

In autism research, neuroinflammation has been characterized by microglial activation and an increase in innate immune cytokines [19]. The proper regulation of inflammatory responses in macrophages, including microglia, relies on transforming growth factor beta (TGF-β) to resolve immune activation [19]. A recent multi-omics study on ASD has found a significant elevation of TGF-β, indicating a dysregulated function of microglia [16]. The logarithmic levels of the most distinctive gut bacteria showed a strong correlation with the concentrations of TGF-β and IL-6 [16]. Moreover, 95 human metabolic pathways exhibited unique expression patterns in the brain tissues of individuals with ASD, which corresponded to microbial pathways and the varied abundance of the host microbiome in relation to autistic traits [16]. These findings suggest a potential interplay of metabolic pathways across the host microbiome and the brain, which could contribute to neuroinflammation and hinder neurodevelopment, resulting in disrupted eating behaviors and unhealthy dietary patterns in ASD. Bidirectionally, several gut metabolites have been identified as potential biomarkers that mediate immune response and influence ASD via the gut–brain axis [46]. Individuals with ASD experienced a notable decrease in Purkinje cells counts, responsible for producing GABA, which may be associated with the increased abundance of *Clostridia* spp. [49]. Additionally, one of the SCFAs, propionate, has been identified as correlating with the development of autistic-like behavior when injected intracerebroventricularly [50]. Overall, microbial metabolites could serve as valuable biomarkers for examining neuroinflammation and behavioral changes in ASD, with implications for identifying intervention targets to improve ASD healthcare.

## 3. Gut Microbiome in Connection with ASD

### 3.1. Microbiota Profile with ASD Clinical Symptoms

Microbiome changes may contribute to the development of autism by producing toxic bacterial byproducts and influencing immune function. Interventions aimed at regulating beneficial gut bacteria could enhance intestinal and mental health, thereby alleviating and reducing autism-related symptoms. However, previous attempts to establish a consistent microbial profile specific to ASD have been inconclusive, likely due to variations in gut microbial signatures influenced by gastrointestinal symptoms and dietary factors, which are associated with restricted food intake and specific domains of autistic symptoms [16,51]. Therefore, how to delineate and group the heterogenous subtypes of individuals with ASD should be addressed. The Lancet Commission has underscored the importance of considering co-occurring challenges, such as impaired intelligence, sensory abnormalities, gastrointestinal symptoms, and sleep disturbances, in addition to the core symptom domains of social communication and restricted/repetitive behaviors and interests [52]. Here, our focus was on delineating distinct microbiota associated with the core clinical symptoms of ASD and other related traits (Figure 2). These findings will facilitate further research into microbiome-targeted interventions aimed at rebalancing gut microbiome composition, thus offering valuable insights for clinical care.

#### 3.1.1. Core Clinical Symptoms

A common finding in the stool samples of children with ASD is a remarkably reduced ratio of *Bacteroidetes* to *Firmicutes* [5]. As a systematic review reported, ASD children exhibited elevated levels of *Akkermansia muciniphila*, *Faecalibacterium prausnitzii*, and *Prevotella* spp., as well as reduced levels of *Escherichia coli*, *Bifidobacterium*, and *Enterococcus* spp. [5]. However, most studies primarily focused on detecting differences in gut microbial composition between children with ASD and neurotypical controls, without establishing correlations between the core symptoms of autism and specific taxa.

A review consisting of 14 studies revealed significant associations of behavior disorders with the relative abundance of *Desulfovibrio*, *Clostridium* spp., and *Bacteroides Vulgatus* [10]. Specifically, fecal samples from individuals with ASD exhibited a higher richness of *Clostridium* spp. Compared to healthy controls, *C. perfrigens* is the most prevalent species [31]. The positive correlation between *Clostridium* spp. and Childhood Autism Rating Scale (CARS) scores highlighted the importance of this bacterial species in gut dysbiosis. Intriguing results also demonstrated a trend of elevated *Desulfovibrio* spp. in ASD children, which was strongly associated with autistic severity, specifically in the restricted/repetitive behavior subscale [53]. In contrast, the ratio of *Bacteroidetes/Firmicutes* exhibited a slightly negative correlation with autistic severity [53]. Similarly, a decreased *Bacteroidetes/Firmicutes* ratio observed in individuals with the highest levels of microbial imbalance exhibited significant associations with autistic symptoms (as measured by the Autism Spectrum Quotient-10) and overall psychopathology (as measured by the Strengths and Difficulties Questionnaire) [54]. It is noteworthy that the identified relationships between microbiota profiles and ASD clinical symptoms were further linked to predicted functional pathways, such as elevated levels of amino acid, lipid, and energy metabolism and digestive system-related functions [7]. By utilizing constrained ordination analysis and the Mantel test, Chen et al. [7] first demonstrated a substantial correlation between microbial community and the Child Behavior Checklist (CBCL) scores in individuals with ASD but not those without ASD. The specific taxa included *Bifidobacterium longum* subsp. *longum*, *Bacteroides species (B. lebeius*, *B. stercoris*, *B. fragilis*, and *B. coprocola*), *Clostridiales bacterium*, *Ruminococcus*, and *Parabacteroides merdae* [7]. Additionally, for the SRS subscales, significant correlations were found with *Bacteroides plebeius*, *B. fragilis*, *B. stercoris*, *Bifidobacterium longum* subsp. *longum*, *Ruminococcus*, *Parabacteroides merdae*, and *Megasphaera micronuciformis* [7].

Noteworthy is the extent to which the associations between gut microbiota alterations and improvements in autistic symptoms after intervention provide more implications for a microbiota-based therapeutic approach to ASD. Yang et al. [55] first evaluated the effects of sulforaphane on children with ASD by correlating the changes in gut microbiota with the improvements of the Ohio State University (OSU) Autism Rating Scale under a 12-week intervention period. They found a positive correlation between the levels of *Atopobium* and unidentified*_Xanthomonadaceae* with the severity of general autistic symptoms [55]. Conversely, a decrease in certain genera, including *Actinomyces*, unidentified_*Erysipelotrichaceae*, *Haemophilus*, *Chelativorans*, and unidentified_*Coriobacteriaceae*, was associated with improved outcomes [55]. Furthermore, autism rating subscales, such as social interaction, repetitive/ritualistic behaviors, and verbal/non-verbal communication, also showed correlations with specific microbial genera [55]. More intriguingly, under a combined intervention of oxytocin and probiotics in preschoolers, Kong et al. [56] constructed a correlation framework on the gut microbiome and SRS subscales. *Eubacterium hallii* exhibited the most pronounced negative correlation with the SRS cognition subscale [56]. Moreover, a significant reduction in this species following combination treatment was positively associated with the SRS cognition subscale at baseline [56]. *Lachnospiraceae* (uncultured) showed a negative correlation with ABC inappropriate speech at baseline [56]. Other bacterial taxa alterations, such as *Rikenelaceae*, *Alistipes*, *Christensenellaceae* R7, and *Ruminococcaceae* UCG-002, showed positive correlations with certain baseline scores, where the abundances of *Rikenelaceae* and *Alistipes* were also significantly correlated with SRS motivation at baseline [56]. However, *Christensenellaceae* R7 and *Ruminococcaceae* UCG-002 were exclusively identified after the combination therapy [56].

#### 3.1.2. Other Related Traits

To determine the gut microbiota’s role in reducing autistic behaviors and related GI problems, Nogay et al. [10] systemically summarized that increased growth of *Clostridium histolyticum*, *C. perfringens*, and *Sutterella*, a higher *Escherichia/Shigella* ratio, and a lower *Bacteroidetes/Firmicutes* ratio were associated with GI issues. To be specific, *C. histolyticum*, which acts as toxin-producer contributing to gut inflammation, has shown a significant correlation with GI symptoms in ASD [57]. Children experiencing more severe GI symptoms exhibited reduced levels of *Desulfovibrio* and *Clostridium* bacteria, and a lower *Bacteroidetes/Firmicutes* ratio [53]. Among children with ASD, there existed significant relationships between higher levels of *Bacteroidetes*, *Actinobacteria*, *Proteobacteria*, and *Verrucomicrobiae*, as well as a lower growth rate of *Firmicutes*, with GI problems [33]. Similarly, a higher *Ruminococcus torques* level was predominantly associated with GI problems in ASD children [58], while the negative associations of the levels of *Prevotellacea*, *Bacteroidaceae*, *Prevotellacea*, *Clostridiaceae*, and *Lachnospiraceae* with GI problems did not reach statistical significance [63]. Furthermore, following an 8-week course of probiotics treatment (*Lactobacillus* and *Bifidobacterium)*, a significant positive correlation was observed between *Lactobacillus* and GI score as assessed by the Pediatric Quality of Life Inventory [59]. Given the close connection of constipation and GI distress, an association was also observed between constipation and increased abundance of *Escherichia*, *Shigella*, and *Clostridium* cluster XVIII, alongside decreased levels of *Gemmiger* and *Ruminococcus* in ASD [6]. However, the association between GI problems and taxonomy relative abundance was controversial, as no specific microbiota was associated with GI symptoms after adjusting for age, body mass index, physical health conditions, methylphenidate use, and medical history of ADHD and GI symptoms [7]. This suggests that GI issues may not directly participate in the association between microbiota and the severity of ASD.

Based on a well-designed multi-omic cohort study, Morton et al. [16] uncovered an ASD-related omics framework along the gut–brain axis that was associated with specific phenotypes of ASD. This framework was characterized by distinct metabolic patterns primarily influenced by microbial species, including *Desulfovibrio*, *Prevotella*, *Bacteroides*, and *Bifidobacterium*, highlighting the connections among gut microbiota, metabolic profiles, and ASD. Furthermore, the most distinct microbial taxa (*Prevotella*, *Bacteroides*, and *Bifidobacterium*) exhibited a strong negative correlation with the changes in inflammatory cytokines [16]. Regarding immune function as reflected by allergy/autoimmunity scores, Kong et al. [64] first identified positive correlations of *butyricimonas* abundance and the ratio of *Bacteroidetes/Firmicutes* with immune dysfunction. Notably, elevated levels of *butyricimonas* and *paraprevotella* were predominantly associated with more severe dietary habit and eating slowness, respectively [64]. By the same token, in children with co-occurring ASD and GI symptoms, multiple proinflammatory cytokines were found to have significant correlations with *Clostridiales* bacteria (*Clostridium lituseburense*, *Terrisporobacter* spp., *Lachnoclostridium bolteae*, and *Clostridium hathewayi*), which were associated with GI distress and mucosal levels of tryptophan and serotonin as well [60]. Based on the observed dysfunction of carbohydrate digestion and disaccharidase deficiency in individuals with ASD and GI symptoms, the abundance of *Clostridium* also showed a positive correlation with disaccharidase activity [65].

Attention-deficit/hyperactivity disorder (ADHD), recognized as a neurodevelopmental disorder, has also shown a close connection with ASD. Using the Mendelian Randomization method, Wang et al. [61] discovered that higher abundances of *Eubacteriu hallii* and *Ruminococcaceae* UCG013, and lower levels of *Butyricicoccus*, *Roseburia*, *Desulfovibrio*, *Lachnospiraceae* NC2004 group, *Romboutsia* genera, and *Oxalobacteraceae* family, were significantly associated with the risk of ADHD. Further investigations into the microbiota–gut–brain axis are warranted to explore the correlation between alterations in specific bacterial abundances, bacterial metabolites, and ASD-related behavioral changes, aiming to delve into the mechanisms underpinning autism pathology.

### 3.2. Microbiota Profile with Atypical Eating Behaviors and Nutritional Status

While there is evidence of a correlation between autistic traits in early childhood and poor diet quality later in life, the direct link between diet quality and autistic traits remains elusive [16]. For instance, a reverse association study showed that fiber-rich diets were associated with specific strains of *Prevotella copri* involved in carbohydrate-breakdown [66], which challenges the understanding of whether gut microbiota is the culprit or the result of selective eating and other disordered eating behaviors in ASD.

Of note, a previous study has shown correlations between gut *Butyricimonas* abundance, eating habits, and allergy/immune functions, suggesting an interrelationship between microbial configuration and nutritional status [64]. In terms of identified dietary pattern (DP) in children with ASD, DP1, characterized by a diverse range of healthy food groups (vegetables, fruit, nuts, grains, dairy, etc.) was found to be associated with altered species diversity, and the abundance of *Erysipelotricaceae*, *Clostridiaceae Clostridium*, and *Oscillospira* bacteria [67]. On the other side, DP2, characterized by the consumption of unhealthy food (fried foods, snacks, condiments, etc.), was associated with abundance changes in *Clostridiaceae Clostridium* and *Oscillospira* [67]. Evidence showed that ASD children with food selectivity displayed higher gut microbiota diversity at the phylum level, with increased presence of *Enterobacteriaceae*, *Proteobacteria Salmonella*, *P. Escherichia/P. Shigella*, and *Clostridium* XIVa genera compared to non-picky eaters [68]. Moreover, Yap et al. [17] observed a notable positive correlation between dietary diversity and taxonomic diversity, while there were minimal associations between ASD diagnosis and gut microbiome. Of note, this research provided a novel diagram on the mediation role of dietary preference, where restricted interests in ASD were linked to a less varied diet, in turn, associated with reduced microbial diversity [17].

Other than gut bacteria, fungal commensals, which can thrive following antibiotic use, also potentially disrupt microbiota reestablishment and contribute to dysbiosis. An abnormal gut fungal community profile with higher levels of *Saccharomyces cerevisiae* and lower levels of *Aspergillus versicolor* was observed in individuals with ASD [69]. However, current research on gut fungals in ASD remains scarce. Therefore, future research is required to grasp the dynamic functional architecture of autism and elucidate the causal relationships between atypical eating behaviors, dietary preferences, the microbiome, and other omics levels.

## 4. Beyond the “Cure”: Potential of Pre-/Probiotics

Despite the promising potential of pre-/probiotics interventions, recognized as microbiota-directed strategies, to alleviate certain neurodevelopmental disorders, the use of these interventions for the treatment of ASD still lacks solid clinical evidence and recommended dosages. As discussed earlier, the wide range of autistic phenotypes and co-occurring problems make it difficult to determine the optimal pre-/probiotics approach. Beyond solely treating ASD as a single syndrome with a singular underlying mechanism of brain dysfunction, a more productive approach may be to explore the multifaceted nature of this condition. Instead of searching for a one-size-fits-all therapeutic strategy, researchers should investigate the potential of pre-/probiotics interventions to comprehensively map gut homeostasis, nutritional status, and immune responses in ASD [52]. This broader perspective could yield valuable insights to guide ASD management strategies. Rather than viewing pre-/probiotics as a direct treatment for ASD symptoms, clinicians and researchers should emphasize understanding the underlying mechanisms by which these interventions modulate the gut microbiome and influence immune activation.

### 4.1. Clinical Evidence on Pre-/Probiotics

In clinical practice, it is crucial to adopt a holistic approach to ASD management that considers the multifaceted interactions between the gut microbiome, dietary factors, immune responses, and nutritional status. Alleviating immune dysregulation has been shown to play a significant role in improving overall health outcomes, surpassing the impact on core symptoms. In this sense, we evaluated previous clinical trials and tabulated the ratios of improved cases to the total number of studies conducted, examining six dimensions: GI symptoms, circulating biomarkers (e.g., inflammatory cytokines, neurotransmitters, and other metabolites), core clinical symptoms (social interaction/communication and restricted/repetitive behaviors), sleep problems, other traits related to ASD, and parents’ ratings of their anxiety/stress and other concerns (Table 1).

In addition to a thorough review of studies sourced from PubMed, Scopus, and Google Scholar (2005–2024), we also utilized the ClinicalTrials.gov database to identify relevant clinical trials with complete and publicly available results. Studies without validated controls were excluded from the summary.

#### 4.1.1. Probiotics

Gut microbial reconfiguration has been targeted as a promising strategy to alleviate inflammatory matters and GI distress, which were frequently accompanied by neurological disorders, such as ASD [70,71]. Many preclinical studies have investigated the efficacy of probiotics in reducing severe symptoms related to ASD, indicating specific effects on various behavioral, mental, and gut health under different probiotic strains [70]. However, gauging the differences in the effects of different probiotic strains and their combinations in clinical settings has still a long way to go. Moreover, some case reports and open-label clinical trials presented weak evidence and inconsistent results [3,53].

Based on the search strategy, five crossover-controlled trials and four randomized controlled trial (RCT) studies were selected. Two registered crossover-controlled trials (NCT03369431 and NCT02903030) did not observe substantial alleviation of symptoms after treatment. In the study (NCT02903030), there was a trend of improved GI issues based on the Pediatric Quality of Life Inventory (PedsQL) measure, but without statistical significance (*p* = 0.096). No significant change was observed in secondary outcomes over 8 weeks. Notwithstanding, the study reported significant improvements in at least one parent-selected target symptoms related to GI issues on 9-point scale. Similarly, a crossover-controlled trial of the probiotic product (VISBIOME) in ASD children showed moderate improvements in PedsQL and Parent-Rated Anxiety Scale for ASD (PRAS-ASD) scores, though these changes did not reach statistical significance [59]. However, the study did report significant improvement in GI discomfort as assessed by parent-selected ratings of target symptoms [59]. In contrast, an earlier report demonstrated that *Lactobacillus plantarum* WCFS1 led to significant improvements in stool consistency and total behavior score, along with increases in *Lactobacilli* and *Enterococci* and a decrease in *Clostridium* cluster XIVa [72]. More recently, a crossover study by Guidetti et al. [73] also found significant improvements in GI symptoms, maladaptive behaviors, and communication skills, although no change in gut microbial diversity was observed under the treatment. Overall, the variation in intervention periods (6–12 weeks), assessment tools for GI symptoms and abnormal behaviors, and the specific probiotic strains used may contribute to the mixed and conflicting results observed across these studies.

A single probiotic (*Lactobacillus plantarum* PS128) RCT intervention study was first reported by Liu et al. [74], which could substantially decrease Swanson, Nolan, and Pelham-IV (SNAP-IV) total score and the score on body and object use subscale of the ABC over a 4-week course of treatment. The treatment group also benefited in reducing anxiety and rule-breaking behaviors based on the CBCL assessment [74]. Notably, subgroup analysis further revealed a modifying effect of age, with more significant improvements in inattention, hyperactivity/impulsivity, and opposition and defiance observed in the younger group (7–12 years old) [74]. Considering the limited intervention duration and potential for prolonged effects of *Lactobacillus Plantarum* PS128, Liu’s team [75] organized the participants into two groups: an early intervention group (E-group, *n* = 41) that received the probiotic for the full 4-week period, and a late group (L-group, *n* = 41) that received the probiotic from the middle stage to the end of week 4. After accounting for differences in fathers’ education, intelligent scores, and fish oil usage at baseline, they observed a significant improvement in the anxious/depressed subscale of the Achenbach System of Empirically Based Assessment (ASEBA) over 2 months in both E-group and L-group. This suggested that the treatment’s effectiveness could be demonstrated and maintained within a timeframe of 2 months [75].

Following combination treatment with eight strains named Vivomixx® for 6 months, those ASD children without GI symptoms experienced a significant decrease in the Autism Diagnostic Observation Schedule (ADOS) total score and improvement in the Social-Affect subdomain [76]. Importantly, GI symptoms, adaptive functioning, and sensory profiles were relieved only in children with co-occurring ASD and GI symptoms [76]. However, when comparing the primary and secondary outcomes between the two arms, ADOS score, plasma biomarkers, and fecal calprotectin were not substantially changed [76]. The absence of improvement in GI symptoms suggested that probiotic administration may have the potential to ameliorate core ASD symptoms independent of GI function mediated by microbiome homeostasis. Given the close connections between neurophysiological measures and clinical or biochemical indicators in ASD development, the exact mechanisms by which probiotics modulate brain function warrant further investigation. In the same cohort, Billeci et al. [77] recently applied electroencephalography (EEG) to observe a declined power within the frontopolar regions in β and γ bands in ASD children who were administered with Vivomixx® mixed probiotics. Moreover, the probiotics led to an accordant shift in frontal asymmetry and an increase in frontopolar coherence in the same bands, aligned with a positive association between restricted/repetitive behaviors and gamma band activity within the frontopolar area [77].

In addition to interventional research using Vivomixx® probiotics in Italy, Mazzone et al. [78] conducted a pilot RCT to examine the effects of *L. Reuteri* (strains ATCC-PTA-6475 and DSM-17938) on ASD children in the same country. Participants receiving *L. Reuteri* treatment exhibited a decrease in blood soluble CD40L levels and improvements in social communication, though not in social motivation or mental state understanding subdomains [78]. Nonetheless, the connections between social behavior and specific changes in immune profile under *L. Reuteri* treatment warrant further investigation. Notably, the most salient improvement was the elevated adaptive social functioning subscale in the treatment arm compared to the placebo [78], again underscoring the potential benefits of single probiotic interventions for alleviating ASD-related symptoms.

Given that the two publications reported findings originated from the same clinical trial [76,77] (NCT02708901), we viewed them as one study to evaluate the overall efficacy of probiotics intervention. It was interesting to find that two dimensions of treatment evaluation (GI symptoms and related traits) manifested significant improvements in over 50% of the results. However, relieves in core clinical symptoms and parental ratings of anxiety/stress were not consistently met across the research.

#### 4.1.2. Prebiotics

So far, investigations on the use of prebiotic supplements in children with ASD remain scarce and limited in scope. Only two documented RCT studies have been performed to explore the potential of prebiotics for ASD management.

As shown in Table 1, Grimaldi et al. [79] randomly divided 30 ASD children into four groups: an elimination diet (mainly casein-free and gluten-free) + beta-galacto-oligosaccharide (B-GOS, Bimuno^®^), an elimination diet + placebo, an unrestricted diet + B-GOS, and an unrestricted diet + placebo, to examine the effect of B-GOS on ASD symptoms and other health outcomes. For ASD children who followed the 6-week integrated intervention of a restricted diet and B-GOS, social skills based on the AQ measure as well as anti-social behavior based on the ATEC showed significant improvements, after adjusting for age and dietary factors [79]. In all ASD children receiving the B-GOS treatment, anti-social behavior improved significantly, accompanied by an increase in the relative abundance of *Lachnospiraceae*. However, only 23% experienced sleep improvement, and there was no substantial alleviation in GI symptoms [79].

Moreover, it is worth mentioning a pilot trial conducted by Raghavan and colleagues, which explored the roles of β-1,3/1,6-glucan on autistic-like behaviors, sleep problems, and gut microbial structures [80,81,82]. As expected, more ASD children experienced significant improvement in the CARS total score when combined with glucan treatment (*p* = 0.034), particularly in the irritability/anger and sleep improvement subscales [80]. The glucan intervention also significantly improved sleep quality, concurred with an increase in melatonin levels at the endpoint [81]. The gut microbial composition was altered following the treatment, with decreased levels of *Escherichia coli*, *Akkermansia muciniphila*, *Blautia* spp., *Coprobacillus* spp., and *Clostridium bolteae*, and increased levels of *Faecalibacterium prausnitzii* and *Prevotella copri*, reflecting a more favourable gut community [82].

As three research articles belonged to the registered study (India: CTRICTRI/2020/10/028322), thereby a total of two RCT trials have investigated the effect of prebiotics for now. It was remarkable to find that the prebiotics treatments manifested great efficacy in improving core clinical symptoms and related symptoms (with two out of two studies showing positive outcomes), despite the quite limited research in this field.

#### 4.1.3. Symbiotic and Combined Treatments

To develop a comprehensive ASD healthcare system, high-priority nutrition-related challenges, autistic clinical phenotypes, and gut–brain dysfunction should be addressed not only through pre-/probiotics treatment, but also through the integration of behavioral training, educational interventions, and parental practices. Notably, a 4-week Applied Behavior Analysis (ABA) training program accompanied by probiotics treatment demonstrated a significant decrease in ATEC total score in ASD children, particularly in the domains of Health/Physical/Behavior, Speech/Language Communication, and Sociability, Sensory/Cognitive Awareness [83]. When subgrouping the treatment group based on the presence or absence of GI problems, probiotics showed greater effectiveness among those who did not initially experience GI discomfort [83]. Additionally, GI problems were significantly reverted in the treatment group compared to the placebo [83].

By the same token, Li et al. [84] administrated a combination of ABA and probiotics to examine their synergistic effects in clinical healthcare of autism [84]. Without surprise, they observed a notable reduction in the ATEC total score and all its subscales. The increased levels of *Bifidobacterium*, *Lactobacillus*, *Coprobacillus*, *Ruminococcus*, *Prevotella*, and *Blautia*, along with decreased levels of *Shigella* and *Clostridium*, indicated a shift towards a more favorable gut microbial construction [84].

Following a probiotics and fructo-oligosaccharide combination therapy, there was an increase in favourable bacteria (*Bifidobacteriales* and *Bifidobacterium longum*), along with a decrease in *Clostridium* (potentially detrimental bacteria) [85]. This intervention resulted in a significant reduction in ATEC total score and GI issues, with improvements in Sociability and Speech/language/communication subscales observed as early as day 60 and maintained at the endpoint (day 108) [85]. Moreover, there was a decrease in serotonin levels and an increase in homo-vanillic acid, indicating improvements in the hyper-serotonergic state and dopamine metabolism disorder, respectively [85]. Certain blood metabolites deficient in ASD children at baseline, such as L-tryptophan, 5-HT, and 5-hydroxyindoleacetic acid, were also reversed to normal ranges after the intervention.

The combined treatment of probiotics and oxytocin demonstrated a significant improvement in the Clinical Global Impressions (CGI) score only, along with a trend towards improvement in the total score of the ABC, stereotyped behavior subscale, and SRS cognition domain [56]. Although no substantial changes in gut microbiome diversity, GI symptoms, and plasma biomarkers were observed, there was a trend towards normalization of ASD-related bacteria genera and predicted functional profile [56].

Other than these RCT studies, a recent crossover pilot trial, incorporating *L. reuteri* with dextran and maltose, has demonstrated that the combined treatment exhibited significant enhancements in adaptive behavior and social preference, assessed by Vineland-3 scale and eye tracking task [86]. Although no substantial changes were observed in the biomarkers tested, including plasma vasopressin, inflammatory markers, stool lactoferrin, and calprotectin [86], further larger-scale study should be conducted.

It was noteworthy that the core clinical symptoms and related symptoms exhibited a relatively high proportion of expected outcomes (up to 60% across the five studies). Meanwhile, improvements in GI symptoms and sleep issues manifested in most of the research on symbiotic and combined interventions.

Altogether, the efficacy of pre-/probiotics treatment showed benefits across multidimensional assessments of health status in children with ASD, except for parental ratings, suggesting a relatively weak effect on parental concerns and anxiety/stress. To be specific, compared to core clinical symptoms, pre-/probiotics demonstrated the most effective results on GI symptoms and other traits related to ASD, with 67% and 80% of the studies reporting positive outcomes, respectively. This, in part, highlights the importance of focusing on general gut health and the relief of ASD comorbidities, rather than barely targeting the core clinical symptoms, in ASD management. In line with our findings, an up to date meta-analysis involving 302 ASD children from the six RCT studies indicated that probiotics could efficiently alleviate GI symptoms rather than ASD core clinical phenotypes [87]. The conclusions drawn from our review may be subject to bias due to several limitations, including non-standardized measures for autistic symptoms, population heterogeneity (i.e., age, medical status, eating behaviors, lifestyle, and the severity of ASD), variations in intervention periods and dosages, as well as limited sample sizes. Nevertheless, we scrutinized all the studies and CONCORT information according to the clinical registered identities to ensure the quality and randomized controlled nature of the research.

### 4.2. Potential Fecal Microbiota Transplantation (FMT) in ASD

For now, the administration of FMT in ASD remains as a new area with no reported RCT. Two retrospective studies on the effects of washed microbiota transplantation among children with ASD suggested significant relief in core symptoms, abnormal behaviors, GI discomfort, and sleep disorders [88,89]. However, the study population all experienced probiotics supplementation previously. In addition, two open-label studies performed by Kang et al. [90] and Li et al. [91] have demonstrated that FMT could effectively improve ASD symptoms, and GI discomfort, and regulate neurotransmitters to normal ranges. Accordingly, Kang et al. [92] conducted a follow-up study to evaluate the long-term efficacy of FMT. Upon reanalyzing the original data, seven out of eight genera, including *Bacteroides_E intestinalis*, *Muribaculum sp002492595*, *Prevotella sp003447235*, *Sutterella wadsworthensis_A*, *Desulfovibrio_R_piger_A*, *Coprobacter_secundus*, and *Parabacteroides_B goldsteini*, were enriched throughout the two years of FMT. These taxa possess a potentially wide functional diversity and have been associated with beneficial immunomodulatory properties [9].

While basic biological research cannot definitively answer the casual relationship or the etiology of autistic traits, evidence supports the rationale behind pre-/probiotic interventions in altering the microbiota and improving ASD-related symptoms. Co-treatment involving personalized diet and FMT is believed to enhance health outcomes more effectively in individuals with ASD by maintaining long-term homeostasis of the gut microbiome and immune states.

**Table 1 nutrients-16-02382-t001:** Clinical studies on the potential of pre-/probiotics or combination treatments from different perspectives in autism healthcare.

Methods	Pre-/Probiotics	Characteristics	Measures	Gut Microbial Changes	GI Issues	Biomarkers	Core Symptoms	Sleep	Related Traits	Parents’ Rating	Ref.
**Probiotics**											
Crossover-controlled	*Lactobacillus plantarum* WCFS1	17 ASD children; 6-week period; Britain	Stool consistency, social behavioral score, fecal sequencing	Increases in *Lactobacilli* and *Enterococci* and decrease in *Clostridium* cluster XIVa.	√	-	√	-	-	-	[72]
Crossover-controlled	VISBIOME: 1 strain of *Streptococcus thermophiles*, 3 strains of *Bifidobacteria*, and 4 strains of *Lactobacilli*	10 ASD children; 8-week period and 3-week washout; USA	GI module of PedsQL, PRAS-ASD, Target Symptom Rating, ABC, SRS, fecal sequencing	No alteration in microbiota composition.	√	-	×	-	×	×	[59]
RCT	*Lactobacillus plantarum* PS128	39 ASD children: probiotic, 41 ASD children: placebo; 4-week period; Taiwan (China)	CGI-I, ABC-T, SRS, SNAP-IV	-	-	-	√	-	√	-	[74]
RCT	*Lactobacillus Plantarum* PS128	41 ASD children: early treat with PS128 (for 4 weeks), 41 ASD children: late treat with PS128 (start from the middle course until week 4); Taiwan (China)	ADHDT, ASEBA	-	-	-	×	-	√	-	[75]
RCT	Vivomixx®: *Streptococcus thermophilus*, *Bifidobacterium* (*B. breve*, *B. longum*, *B. infantis*), and *Lactobacillus* (*L. acidophilus*, *L. plantarum*, *L. paracasei*, *L. delbrueckii* subsp. *Bulgaricus*)	42 ASD children (14 GI and 28 NGI): probiotic, 43 ASD children (16 GI and 27 NGI: placebo; 6-month period; Italy	GSI, ADOS-CSS, VABS-II, SCQ, SP, RBS-R, CBCL, PSI, plasma Leptin, TNF-α, IL-6, PAI-1, EEG	-	√	×	√	-	√	×	[76,77]
Crossover-controlled	*Limosilactobacillus fermentum* LF10, *Ligilactobacillus salivarius* LS03, *Lactiplantibacillus plantarum*, and 5 trains of *Bifidobacterium longum*	61 ASD children; 3-month period and 2-month washout; Italy	GSI, PSI, VABS, ASRS, fecal sequencing	Beta diversity altered, and *S. thermophilus*, *B. longum*, *Lfermentum*, and *L. salivarius* showed high abundances in the treatment group.	√	-	×	-	√	√	[73]
RCT	*Lactiplantibacillus* Reuteri	21 ASD children: probiotics, 22 ASD children: placebo; 6-month period; Italy	ADOS, GSI, RBS-R, ABAS-II, PSI, SRS, CBCL, blood CD40L and immune response, fecal sequencing	No widespread changes in the diversity and composition of gut microbiome.	×	√	√	-	√	×	[78]
Crossover-controlled	VISBIOME: 1 strain of *Streptococcus thermophiles*, 3 strains of *Bifidobacteria*, and 4 strains of *Lactobacilli*	13 ASD children; 8-week period and 3-week washout; USA	GI module of PedsQL, Target Symptom Rating, ABC, SRS, CSHQ, PSI	-	√	-	×	×	×	×	NCT02903030
Crossover-controlled	Vivomixx®: *Streptococcus thermophilus*, *Bifidobacterium* (*B. breve*, *B. longum*, *B. infantis*), and *Lactobacillus* (*L. acidophilus*, *L. plantarum*, *L. paracasei*, *L. delbrueckii* subsp. *Bulgaricus*)	69 ASD children; 12-week period and 4-week washout; Bratish	ATEC, GIH, ABC, APSI	-	×	-	×	-	×	×	NCT03369431
**Prebiotics**											
RCT	Bimuno® ^1^	30 ASD children, in which URD+B-GOS (*n* = 7), URD+placebo (*n* = 7), RD+B-GOS (*n* = 6), RD+placebo (*n* = 6); 6-week period; Britain	GI symptoms, stool consistency, ATEC, AQ, SCAS-P, sleep diary, fecal sequencing	An increase in *Lachnospiraceae* family.	×	-	√	×	√	-	[79]
RCT	β-1,3/1,6-glucan	12 ASD children: conventional treatment combined with the glucan, 6 ASD children: conventional treatment; 90-day period; India	CARS, CSHQ, plasma alpha-synuclein, serum melatonin, fecal sequencing	A shift towards a healthier microbial composition with decreased *Escherichia coli*, *Akkermansia muciniphila*, *Blautia*, *Coprobacillus*, and *Clostridium bolteae.*	√	√	√	√	√	-	[80,81,82]
**Symbiotic and combined treatments**											
RCT	6 bacteria (the strain not shown)	37 ASD children: a combination of ABA training and probiotic, 28 ASD children: ABA training only; 4-week period; China	ATEC, GI score	-	√	-	√	√	√	-	[83]
RCT	*Bifidobacterium infantis* Bi-26, *Lactobacillus rhamnosus* HN001, *Bifidobacterium lactis* BL-04, and *Lactobacillus paracasei* LPC-37	16 ASD children: a combination of fructo-oligosaccharide and probiotic, 10 ASD children: placebo; 108-day period; China	ATEC, GSI, fecal sequencing	A shift towards a healthier microbial composition with increased *Bifidobacteriales* and *B. longum*, and reduced *Clostridium*.	√	√	√	-	√	-	[85]
RCT	*Bifidobacterium longiformis*, *Lactobacillus acidophilus*, *Enterococcus faecalis*	21 ASD children: a combination of ABA training and probiotic, 20 ASD children: ABA training only; 3-month period; China	GSI, ATEC, fecal sequencing	A shift towards a healthier microbial composition with increased *Bifidobacterium*, *Lactobacillus*, *Coprobacillus*, *Ruminococcus*, *Prevotella*, and *Blautia*, and decreased *Shigella* and *Clostridium*.	-	-	√	-	√	-	[84]
RCT	*Lactobacillus plantarum* PS128	18 ASD children: probiotic, 17 ASD children: placebo; 28-week period and oxytocin treatment for all groups starting on 16 weeks; USA	SRS, ABC, plasma OXT, IL-1β, CGI, GSI, fecal sequencing	A trend towards the normalization of ASD-related bacteria genera and the predicted functional profile.	×	×	×	-	√	-	[56]
Crossover-controlled	A combination of *L. reuteri*, *dextran*, *and maltose*	15 ASD children; 28-day period and 14-day washout; USA	ADOS, SCR, WASI-II, ABC, CGI, RBANS, VABS, attentional performance, neurophysiology measures, plasma OXT, and serum inflammatory factors	-	-	×	×	-	√	-	[86]
**Efficacy evaluations**											
**Cases with significant improvements**			**Probiotics**		5/7	1/2	4/9	0/1	5/8	1/6	
**Percentages (%)**					71%	50%	44%	0	63%	17%	
**Cases with significant improvements**			**Prebiotics**		1/2	1/1	2/2	1/2	2/2	-	
**Percentages (%)**					50%	100%	100%	50%	100%	-	
**Cases with significant improvements**			**Symbiotic and combined treatments**		2/3	1/3	3/5	1/1	5/5	-	
**Percentages (%)**					67%	33%	60%	100%	100%	-	
**Cases with significant improvements**			**Overall**		8/12	3/6	9/16	2/4	12/15	1/6	
**Percentages (%)**					67%	50%	56%	50%	80%	17%	

GI: Gastrointestinal; Biomarkers: Improvements in inflammatory cytokines, neurotransmitters, and other metabolites from blood/fecal/mucosal samples; Related traits: Improvements in other ASD-related manifestations, such as anxious/depressed states, somatic distress, attention deficit, aggressive behavior, and internalizing/externalizing issues, apart from core symptoms; Parents’ rating: Improvements in parental reported problems, such as parental anxiety, stress, and parental concerns on their children with severe autistic traits. For each clinical trial, any significant improvements observed in the intervention group were indicated with a checkmark (√) across the six assessed domains: GI issues, Biomarkers, Core Symptoms, Sleep, Related Traits, Parents’ Rating related traits. Conversely, a cross mark (×) was used to denote no significant changes. A hyphen (-) was expressed when the study did not conduct certain assessment. ASD: Autism spectrum disorder; PedsQL: Pediatric Quality of Life Inventory; PRAS-ASD: Parent-Rated Anxiety Scale for autism spectrum disorder; ABC: Aberrant Behavior Checklist; SRS: Social Responsiveness Scale; CGI: Clinical Global Impression; ABC-T: Autism Behavior Checklist-Taiwan version; SNAP-IV: Swanson, Nolan, and Pelham-IV-Taiwan version; ADHDT: Attention-Deficit/Hyperactivity Disorder Test; ASEBA: Achenbach System of Empirically Based Assessment; NGI: Non-GI symptoms; GSI: Gastrointestinal Severity Index; ADOS-CSS: Total Autism Diagnostic Observation Schedule-Calibrated Severity Score; VABS-II: the Vineland Adaptive Behavior Scales-II; SCQ: Social Communication Questionnaire; SP: Sensory Profile; RBS-R: Repetitive Behavior Scale-Revised; CBCL: emotional, behavioral, and social problems screening through the Child Behavior Checklist; PSI: Parenting Stress Index; TNF-α: Tumor Necrosis Factor-alpha; IL-6: Interleukin-6; PAI-1: Plasminogen Activator Inhibitor-1; EEG: Electroencephalography; VABS: Vineland Adaptive Behavior Scale; ASRS: Autism Spectrum Rating Scale; ABAS-II: Adaptive Behavior Assessment System–Second Edition; CSHQ: Children’s Sleep Habits Questionnaire; ATEC: Autism Treatment Evaluation Checklist; GIH: Gastrointestinal History survey; APSI: Autism Parenting Stress Index; URD: Unrestricted diet; RD: Restricted diet; B-GOS: Beta-galacto-oligosaccharide; AQ: Autism Quotient; SCAS-P: Spence’s Children Anxiety Scale-Parent version; CARS: Childhood Autism Rating Scale; ABA: Applied Behavior Analysis; OXT: Oxytocin; WASI-II, Wechsler Abbreviated Scale of Intelligence, 2nd Edition; RBANS, Repeatable battery for assessment of neuropsychological status; VABS, Vineland Adaptive Behavior Scales, 3rd edition. The checkmark and cross symbols indicate significant and insignificant improvements, respectively. The dash symbol represents no data reported in the study. Tick and cross represent significant and insignificant improvement for certain perspective, while dash represents not reported in the study. ^1^ Bimuno^®^ contains a composition of galacto-oligosaccharides (B-GOS). While the intervention combined B-GOS with a restricted diet, the focus of the study revolved around the clinical improvements in groups with and without B-GOS administration. Therefore, this study was included as prebiotics intervention trial.

## 5. Limitations and Future Strategies

The impact of eating disorders and gut dysbiosis on the concurrent functional impairments in the neurological system emphasizes the importance of incorporating pre/probiotics with dietary interventions for the treatment of ASD. Altered preferences arising from sensory aversions and prior GI issues in ASD may contribute to compromised nutritional status and gut dysbiosis, perpetuating a vicious cycle involving immune activation and exacerbation of behavioral abnormalities [93]. As proposed by Tomova et al. [68], modifying the dietary habits of ASD children could correct their gut microbiota composition, particularly for those species exhibited links with food selectivity and a preference for healthy dietary patterns. However, determining the temporal relationships among atypical eating behaviors, nutritional insufficiency, and gut microbiome-mediated immune dysregulation in ASD presents a significant challenge. This complicates our understanding of the gut–brain interactions, when evaluating the effects of pre/probiotic interventions.

Another significant gap in this field is the broad spectrum of ASD phenotypes, ranging from mild to severe, and the lack of well-matched controls in previous gut microbiome profiling research. To date, the diversity of autism-related symptoms and the variability of assessment tools present a clinical challenge in standardizing and normalizing the effectiveness of behavioral improvements across different interventions [94]. It is crucial to identify and address co-occurring conditions to adopt a personalized approach to ASD management. Moreover, distinguishing between metabolites originating from endogenous factors (microbes, immunity) and exogenous factors (diet, drugs) is challenging in complex organisms, due to the lack of standardized methods. This hinders research on gut microbiome-directed care and understanding the crosstalk between gut–brain health and immune response in ASD. The ongoing challenge still lies in determining the most effective combination of pre-/probiotics for specific populations, considering factors such as timing and duration, to meet their individual needs.

Moreover, to better decipher the connection between the gut and neurodevelopment within the complex food system, it is necessary to conduct multidisciplinary research that encompasses diverse fields, like computing and mathematic. Nutritional status and gut–brain health in ASD children are significant concerns that should be addressed by both their parents/caregivers and interdisciplinary health professionals involved in their care, including developmental behavioral pediatricians, occupational therapists, registered dietitians, certified behavior analysts, and speech-language pathologists [95]. In addition to the use of pre-/probiotics in ASD clinical research, exploring other gut microbiome-targeted approaches by incorporating digestive enzyme or fermented food supplementation, could shed light on addressing autistic traits and nutritional problems [8,44]. Furthermore, the development of specific clinical guidelines regarding the optimal dosage, formulation, and duration of pre-/probiotic interventions holds promise for the future.

Finally, the goal of ASD treatment may not be a “cure”, but rather to reduce symptoms to a point where individuals no longer require extensive long-term support [96]. This objective is particularly relevant for individuals with “profound autism”, who have more severe core symptoms and co-occurring conditions, necessitating lifelong care [52]. While therapeutic research has traditionally focused on core symptoms of ASD, targeting associated symptoms alongside the core deficits could have a more profound impact on improving overall functioning and health outcomes in autism [96]. Therefore, it is essential to follow specific criteria in identifying (1) the most problematic symptoms, (2) the traits most likely to have widespread impacts on other symptoms, (3) the symptoms most likely to respond to treatment, and (4) treatments backed by the strongest evidence [96]. This approach helps in developing effective and personalized interventional strategies for individuals with ASD. In this regard, improving support and resources related to nutrition for ASD individuals and their families requires interdisciplinary research and collaborative efforts, thus achieving individualized management and prognosis.

## 6. Conclusions

The connection between the host and gut microbiota dysbiosis in children with ASD presents numerous challenges in clinical healthcare, resulting in metabolic disturbances, immune activation, nervous system dysfunction, and disordered eating behaviors. Furthermore, the concurrent immune activation and atypical eating behaviors can lead to imbalanced diet and malnutrition, creating a vicious cycle that further exacerbates ASD management. While the core clinical symptoms of ASD remain a primary focus, pre-/probiotic interventions have demonstrated the most effective way in addressing GI issues and other ASD-related traits, with positive outcomes reported in 67% and 80% of the studies, respectively. This emphasizes the significance of prioritizing gut health and alleviating comorbidities in ASD management. By targeting the gut microbiome, these interventions offer a promising avenue for reducing immune dysregulation and restoring nutritional status in ASD. Further interdisciplinary research is needed to identify the most effective combinations of pre-/probiotics and establish optimal protocols for their use. Ultimately, this will enable more personalized management and enhance the overall quality of life for children with ASD.

## Figures and Tables

**Figure 1 nutrients-16-02382-f001:**
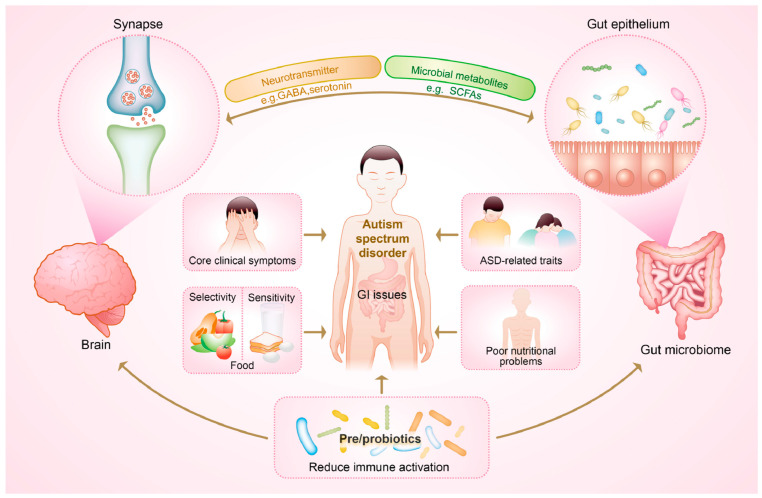
The potential of pre-/probiotics in alleviating gut microbiome-mediated immune dysregulation and improving general health outcomes via the crosstalk between gut microbiota and the brain in children with autism spectrum disorder (ASD). GI, gastrointestinal; GABA, γ-aminobutyric acid; SCFAs, short-chain fatty acids.

**Figure 2 nutrients-16-02382-f002:**
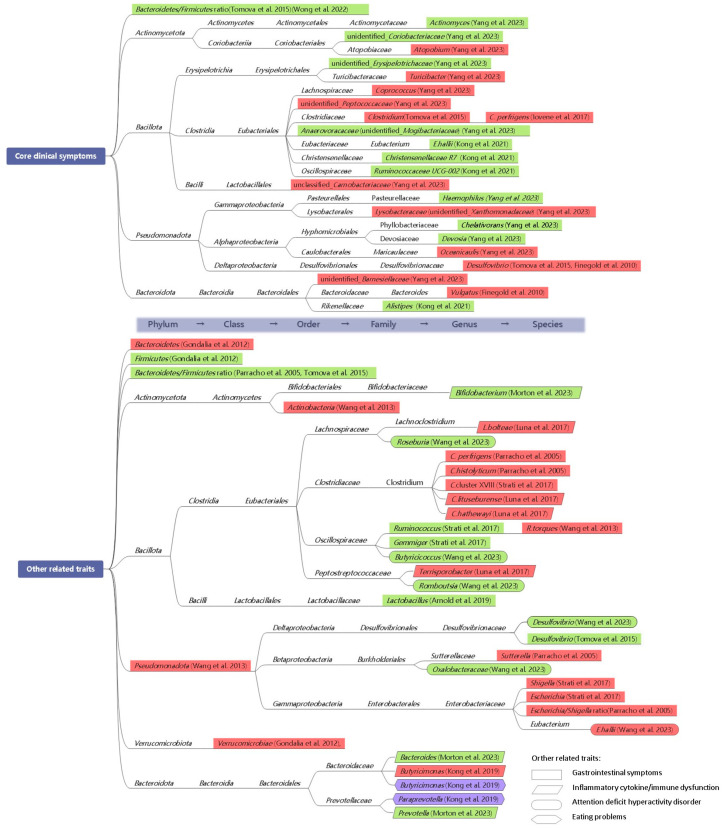
Atlas illustrating the associations among core symptoms of autism spectrum disorder (ASD), other related traits and bacterial taxa identified in children with ASD. Other related traits include gastrointestinal symptoms, inflammatory cytokine/immune dysfunction, and attention deficit hyperactivity disorder, represented in rectangles, parallelograms, and curved boxes, respectively. The positive correlations between eating problems and two species of *Prevotellaceae* are encased in purple rhombuses. Green shading denotes negative linear relationships between the severity of certain symptoms and bacterial taxa, while red indicates positive linear associations observed in individuals with ASD. The taxonomy tree linking these documented microbial taxa is presented on a white background [6,16,31,33,53,54,55,56,57,58,59,60,61,62].

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
