# Peer review of "Harnessing the Gut Microbiome: To What Extent Can Pre-/Probiotics Alleviate Immune Activation in Autism Spectrum Disorder?"

_nutrients, 2024, doi:10.3390/nu16152382_

Round 1

Reviewer 1 Report

Comments and Suggestions for Authors

The review is highly interesting. The paper is well-written and carefully revised by the Authors.

Overall, I highly recommend publishing the paper in its present form. Your work is a significant contribution to the field and I am confident it will be well-received by readers.

However, I have some suggestions for improving the quality.

The most crucial suggestion is to include the latest meta-analysis in the studies that assess the effect of probiotics supplementation on ASD. This will provide a comprehensive overview of the current research landscape and highlight the significance of your work. https://link.springer.com/article/10.1186/s13052-024-01692-z

Also study not included in meta-analysis https://www.cell.com/cell-host-microbe/fulltext/S1931-3128(23)00471-7

Some less important issues

I suggest to avoid presented the results of statistical analysis from other studies, f.e., The logarithmic levels of the most distinctive gut bacteria showed a strong correlation with the concentrations of TGF-β and IL-6 (TGF-β: R=0.61, P=1.84×10−5 ; IL-6: R=0.73, 287 P=5.74×10-8 )

Only the main findings, without R, p could be shown.

The results presented in Figure 2 are blurry. Please reconsider this visualisation. I suggest a larger font. The presented results could also include more details about symptoms (add more specification on the Figure if applicable, e.g., what type of other-related traits/ symptoms).

In Table 1, in the column 'Pre-/probiotics', it is essential to always mention the specific strains in products such as Vivomix or Visibiome. This will greatly enhance the value of your research and its applicability in the field.

Reviewer 2 Report

Comments and Suggestions for Authors

This article is a review on the pathogenetic role of gut-brain dysfunction in autism spectrum disorder, with emphasis on the potential of pre-/probiotics in the management of ASD and related gastrointestinal problems.

This review is extensive and informative, with nice tables and figures.

The reviewer sees no major issues in this paper, and would advise authors to address several minor issues:

1.    In Figure 1, the dashed-lined box at the bottom should be labeled as “pre-/probiotics”.

2.    In the legend to Figure 2, “triangles” should be “rectangles” (Line 472).

3.    The word “while” seems to be inadequately used (Lines 342-343 and Lines 593-594).

Comments on the Quality of English Language

The quality of English is very high.
